# Olivine-rich achondrites from Vesta and the missing mantle problem

Zoltan Vaci [1,2 ✉], James M. D. Day [3], Marine Paquet [3], Karen Ziegler[1,2], Qing-Zhu Yin [4], Supratim Dey [4], Audrey Miller [4], Carl Agee[1,2], Rainer Bartoschewitz[5] & Andreas Pack[6]

Mantles of rocky planets are dominantly composed of olivine and its high-pressure polymorphs, according to seismic data of Earth's interior, the mineralogy of natural samples, and modelling results. The missing mantle problem represents the paucity of olivine-rich material among meteorite samples and remote observation of asteroids, given how common differentiated planetesimals were in the early Solar System. Here we report the discovery of new olivine-rich meteorites that have asteroidal origins and are related to V-type asteroids or vestoids. Northwest Africa 12217, 12319, and 12562 are dunites and lherzolite cumulates that have siderophile element abundances consistent with origins on highly differentiated asteroidal bodies that experienced core formation, and with trace element and oxygen and chromium isotopic compositions associated with the howardite-eucrite-diogenite meteorites. These meteorites represent a step towards the end of the shortage of olivine-rich material, allowing for full examination of differentiation processes acting on planetesimals in the earliest epoch of the Solar System.

[1] Institute of Meteoritics, University of New Mexico, Albuquerque, NM, USA. [2] Department of Earth and Planetary Sciences, University of New Mexico, Albuquerque, NM, USA. [3] Scripps Institution of Oceanography, University of California San Diego, La Jolla, CA, USA. [4] Department of Earth and Planetary Sciences, University of California Davis, Davis, CA, USA. [5] Bartoschewitz Meteorite Laboratory, Gifhorn, Germany. [6] Georg-August-Universität, Göttingen, Germany. ✉email: zmoney@unm.edu

The mantle compositions of terrestrial planets are well constrained compared with those of small bodies and asteroids, although it is widely held that smaller differentiated planetary bodies with metallic cores also have olivine-dominated silicate mantles. For example, the asteroid 4 Vesta, though only about 500 km in diameter, is suggested to have undergone core formation and differentiation to form an olivine-rich mantle during a magma ocean phase[1–3], through partial melting[4–6], or through serial magmatism and fractional crystallization[7,8]. If differentiated bodies were ubiquitous throughout the early Solar System, as planetesimal accretion theory suggests[9,10], then olivine-rich asteroid interiors, and meteorite samples of them, should be quite common.

Based on the currently available meteorites and remote sensing of asteroids, olivine-rich asteroid materials appear to be remarkably sparse, implying a 'missing mantle problem'[11]. Notably, the diversity of iron meteorite types suggests the presence of at least 50 different parent bodies with iron cores and complementary olivine-dominated mantles not observed by remote sensing[12]. In the asteroid belt, objects with spectral signatures suggesting that their compositions are olivine-rich (>80%) are classified as A-type asteroids[13], and these account for <0.16% of all objects larger than 2 km[14]. In the meteorite record, diogenites, thought to represent deep crustal cumulate material from Vesta, are mostly orthopyroxenites with under 40 vol% olivine[15], and only small clasts of dunite have thus far been identified in a mesosiderite breccia[16]. Other olivine-rich asteroid-derived rocks include the brachinites, brachinite-like achondrites (BLA), and ureilites. The brachinites and brachinite-like achondrites contain 68–95 modal% olivine, which is somewhat ferroan ($Fo_{64-74}$)[17,18]. Their nearly chondritic trace element compositions indicate that they are dominantly residues formed by partial melt extraction from a chondritic source[17,19], with some meteorites (ALH 84025, Brachina, EET 99402/99407, and Eagles Nest) possibly representing cumulates[20–22]. Ureilites are also primitive achondrites that are likely to represent samples from a planetary mantle that was catastrophically disrupted before it could fully differentiate[23]. They range in composition from dunite to peridotite (olivine $Fo_{74-97}$)[24], but like brachinites and brachinite-like achondrites, they are not from a fully differentiated asteroid.

Several hypotheses have been put forward to explain the apparently low abundance of olivine-rich mantle material in the Solar System. The most obvious of these is that olivine-rich material is not easily detected in the asteroid belt, and there have not been enough impacts involving this undetected material to bring samples of it to Earth. Such a biasing could be due to space weathering, which could darken the spectra of A-type asteroids and obscure them from optical observation[25,26], or to catastrophic impacts that have systematically destroyed olivine-rich planetary mantles in the early Solar System[27]. Dynamical modeling demonstrates that large impacts primarily occurred in the final stages of planetesimal accretion and were capable of stripping away the silicate portions of early planets, leaving behind Fe-rich cores[28]. The disrupted mantle material from these 'stripping' events must then be removed from the accretion disc, presumably by being incorporated into the terrestrial planets or falling into the Sun. Such a scenario cannot account for the largest asteroids Vesta or Ceres, which retain plentiful silicate materials[29,30].

The paucity of olivine-rich material is potentially due to collisional transport processes. Recent asteroid surveys have concluded that there are no olivine-rich bodies in the asteroid belt above ~2 km in size other than A-type asteroids[14], and therefore, there is no statistically significant remote sensing evidence for widespread differentiation into Fe-rich cores, olivine-rich mantles, and basaltic crusts in the asteroid belt[31]. The distribution of

the A-type asteroids instead supports differentiation of planetesimals in the inner Solar System, where the abundance of aluminum-26 allowed for early enhanced melting of these bodies, followed by later migration and implantation into the asteroid belt. The Nice[32] and Grand Tack[33] models provided dynamical explanations for such early disruptions through large-scale orbital excitations, migrations, and ejections in the early Solar System. Later models[34,35] have suggested that the in situ formation of Jupiter was enough to trigger widespread orbital instabilities in the asteroid belt necessary for both the collisional erosion of early planetesimals and their migration to different parts of the Solar System.

Here we reevaluate the missing mantle problem in light of the discovery of three olivine-rich ultramafic achondrite meteorites. The mineralogy and petrology, stable isotope geochemistry, and trace element geochemistry of these rocks suggest that they are sourced from a fully differentiated parent body that is likely to be Vesta or one of the vestoids. They are igneous cumulates that represent the first olivine-rich mantle samples from a small body that experienced core formation and silicate differentiation, likely in the form of serial magmatism that produced large plutonic bodies that were later disrupted by impact.

## Results

**Petrology.** Northwest Africa (NWA) 12217, 12319, and 12562 are recent meteorite finds containing abundant Mg-rich olivine, and are consistent with a cumulate origin, probably representing mantle materials similar to Vesta or the vestoids. The NWA 12217, 12319, and 12562 meteorites are monomict fragmental breccias predominantly composed of olivine fragments up to 1 cm in size that appear cream-colored to light green in hand sample (Figs. 1a, S1). All three breccia assemblages are crosscut by dark veins. Dunite NWA 12217 is composed of 93 modal% olivine, 4 modal% low and high-Ca pyroxene, minor chromite, FeNi metal, andesine plagioclase, Fe-sulfide, merrillite, and trace alkali feldspar and silica, whereas lherzolites NWA 12319 and NWA 12562 have lesser olivine (85–87 modal%) and higher abundances of low and high-Ca pyroxene (9–11 modal%), and minor chromite, FeNi metal, and plagioclase. Northwest Africa 12319 is also notable for having 3 modal% sulfide. The dark veins identified in all three meteorites are composed of variable amounts of sulfide, chromite, and pyroxene, as well as ubiquitous vermicular symplectites composed of chromite and both low and high-Ca pyroxene. These symplectites appear both as inclusions within veins and on their own. The veins and symplectites are found both along grain boundaries and fully enclosed within olivine grains. Despite being brecciated, all three rocks lack "matrix" material. Olivine grains in NWA 12217 exhibit undulose extinction indicative of shock (Fig. S2). The planar fractures and mosaicism in olivine grains record a maximum shock pressure of 60–65 GPa[36]. However, given the lack of any partial melt veins or pockets, the shock pressures that these meteorites experienced are significantly lower.

All three meteorites have olivine grains and fragments that are forsteritic and contain low CaO (average of $0.06 \pm 0.04$ wt%) and $Cr_2O_3$ (average of $0.07 \pm 0.05$ wt%) relative to olivine in other primitive achondrites[37]. Olivine compositions are generally more variable in the lherzolites NWA 12319 ($Fo_{77-91}$) and NWA 12562 ($Fo_{73-92}$) than in the dunite NWA 12217 ($Fo_{86-93}$). Pyroxene compositions show similar variability in NWA 12217 ($Fs_{3-48}$, $Wo_{1-43}$), NWA 12319 ($Fs_{9-46}$, $Wo_{2-44}$), and NWA 12562 ($Fs_{14-56}$, $Wo_{2-37}$) (Fig. S3). These phases do not show zonation in major or minor element composition. Feldspar is a minor (<1 vol%) but ubiquitous phase in NWA 12217, 12319, and 12562, and is evenly distributed as brecciated fragments in the lherzolites. Feldspar-bearing

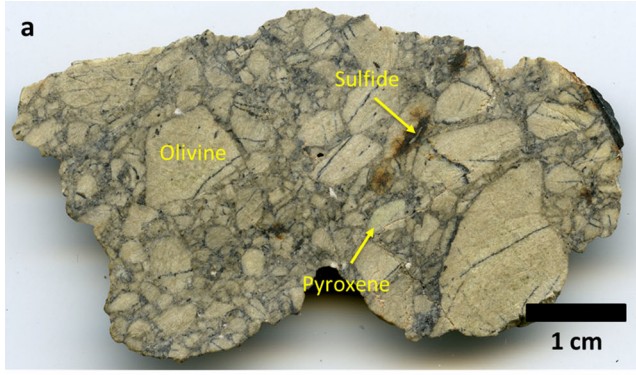

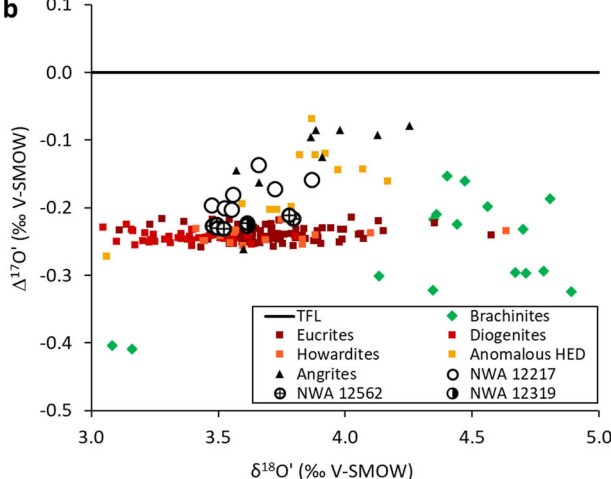

**Fig. 1 Hand sample and oxygen isotopic data. a** Northwest Africa 12217 showing large olivine clasts. The light to dark brown material is terrestrial weathering, typically from oxidation of metal grains and sulfides. Remnant fusion crust is visible in the top right corner of the meteorite. Interstitial areas between grains have been darkened by shock, and they reflect the primary lithology of the meteorites. **b** $\Delta^{17}O$ diagram showing NWA 12217, 12319, and 12562 in relation to other achondrite groups[39,62–64]. The TFL is the mass-dependent terrestrial fractionation line. Errors are smaller than symbols.

fragments are often associated with high-$SiO_2$ (>70 wt%) phases or pure silica. In the dunite NWA 12217, feldspar also occurs as small (~200–300 μm) inclusions within olivine, surrounded by radiating fractures, with compositions ranging from andesine to oligoclase to potassium feldspar (Figs. S4 and S5). No potassium feldspar was identified in NWA 12562 or NWA 12319, but a small high-silica (95 wt%) glassy inclusion was identified in NWA 12562 with 1 wt% $K_2O$. Chromite compositions in all three meteorites show a range in $Al_2O_3$ content of 0 to 25 wt%. Iron-nickel metal in the meteorites ranges in Ni content between 6 and 40 wt%, with Ni/Co between 2 and 192. Chemical zonation is not observed in any of the major or minor phases such that variation in composition is attributed to differences between individual grains.

**Isotopic and trace element geochemistry.** The ultramafic achondrites NWA 12319 and 12562 overlap in oxygen isotope composition as to be virtually indistinguishable, while NWA 12217 is somewhat heavier in $^{17}O$ (overall $\delta^{17}O' = 1.62$ to 1.88; $\delta^{18}O' = 3.48$ to 3.87; $\Delta^{17}O' = -0.230$ to $-0.136$) (Fig. 1b). The chromium isotopic composition of the ultramafic achondrites ranges from $\varepsilon^{54}Cr = -0.65 \pm 0.06$ to $-0.70 \pm 0.07$ (Fig. 2). The coupled $\Delta^{17}O$ and $\varepsilon^{54}Cr$ systematics of NWA 12319 and 12562 plot in the middle of the region of the main group normal HED

meteorites, while NWA 12217 plots slightly above due to its heavier oxygen isotopic composition (Fig. 2b). The isotopic compositions of NWA 12319 and 12562 argue for a heritage from a reservoir similar to normal eucrites and diogenites, while NWA 12217 may have been influenced by mixing with an ordinary chondrite component[38,39]. This could be similar to Moama[38] (a cumulate eucrite), which has a higher $\Delta^{17}O$ but normal $\varepsilon^{54}Cr$, or Sariçiçek, a recently observed howardite fall with lower $\Delta^{17}O$ but normal $\varepsilon^{54}Cr$ that shows signs of mixing with carbonaceous chondrites[40].

Bulk major, minor, and trace element abundances for NWA 12217, 12319, and 12562 reflect the dominance of olivine, in conjunction with a minor feldspar component. Dunite NWA 12217 has a Mg# [molar Mg/(Mg + Fe)] of 91, higher than lherzolites NWA 12319 and NWA 12562 (Mg# = 85 and 83, respectively). The incompatible trace element (ITE) compositions of NWA 12217, 12319, and 12562 are nearly identical in relative abundances (Fig. 3a). The patterns of NWA 12217 and 12562 contain slight enrichments in the light REE but otherwise are almost flat relative to CI chondrites. The variability of the ITE in brachinites makes meaningful comparison with the ultramafic achondrites difficult. The diogenites, by contrast, show similar trends in ITE depletions to NWA 12217, 12319, and 12562.

Highly siderophile element (HSE) concentrations in NWA 12217, 12319, and 12562 are shown in Fig. 3b along with those of the brachinites, diogenites, and bulk silicate Earth and Mars. Abundances of HSE are generally low (<0.002×CI chondrite) with the exceptions of Pd and Re in NWA 12562. All three meteorites have HSE abundances in the intermediate range of the diogenites, with some heterogeneity among aliquots. The osmium isotopic compositions of all three meteorites are almost identical. However, their $^{187}Re/^{188}Os$ ratios vary significantly with four aliquots lying close to a 4.5 Ga Solar System isochron (Fig. S6) whereas one fragment of NWA 12562 has experienced significant Re gain. This combined evidence suggests that the high Re in aliquots of NWA 12562 represent addition from terrestrial weathering, and that the HSE in NWA 12217, 12319, and 12562 are in chondritic-relative abundances. While terrestrial disturbance of Re in meteorites is well documented, disturbance in Pd, especially to the degree noted in NWA 12562, is not[41]. The meteorite also has elevated abundances of Ni, Co, Cu, and Pb, consistent with preferential sampling of sulfide-rich fractions. As veins containing sulfide are found throughout each meteorite (Fig. S7), the elevated Pd in NWA 12562 is consistent with sulfide injection forming veins within the olivine cumulate. It is not possible to accurately determine whether this process occurred immediately after silicate crystallization or perhaps later, as the Re/Os disturbance in the meteorite must be assumed to result from weathering and rusting of sulfides and metals. Not including the additions to NWA 12562, the total measured HSE abundances of all three meteorites are between 1 and 4 ppb, and this is lower than in terrestrial peridotites[41] and much lower than in brachinites and brachinite-like achondrites[17,18].

## Discussion

Further geochemical aspects of the ultramafic achondrites rule out association with other meteorite groups such as angrites and brachinites. The angrites in general appear to be more oxidized than NWA 12217, 12319, and 12562, with analytical and experimental studies suggesting that they crystallized well above the iron-wüstite (IW) buffer[42,43]. The olivine compositions and the presence of Fe-metal in these new meteorites suggest that they were formed in a more reducing environment. A single dunitic angrite, NWA 8535, has been identified[44], and though it also contains Mg-rich olivine ($Fo_{70-88}$), its olivine Fe/Mn is ~90, more

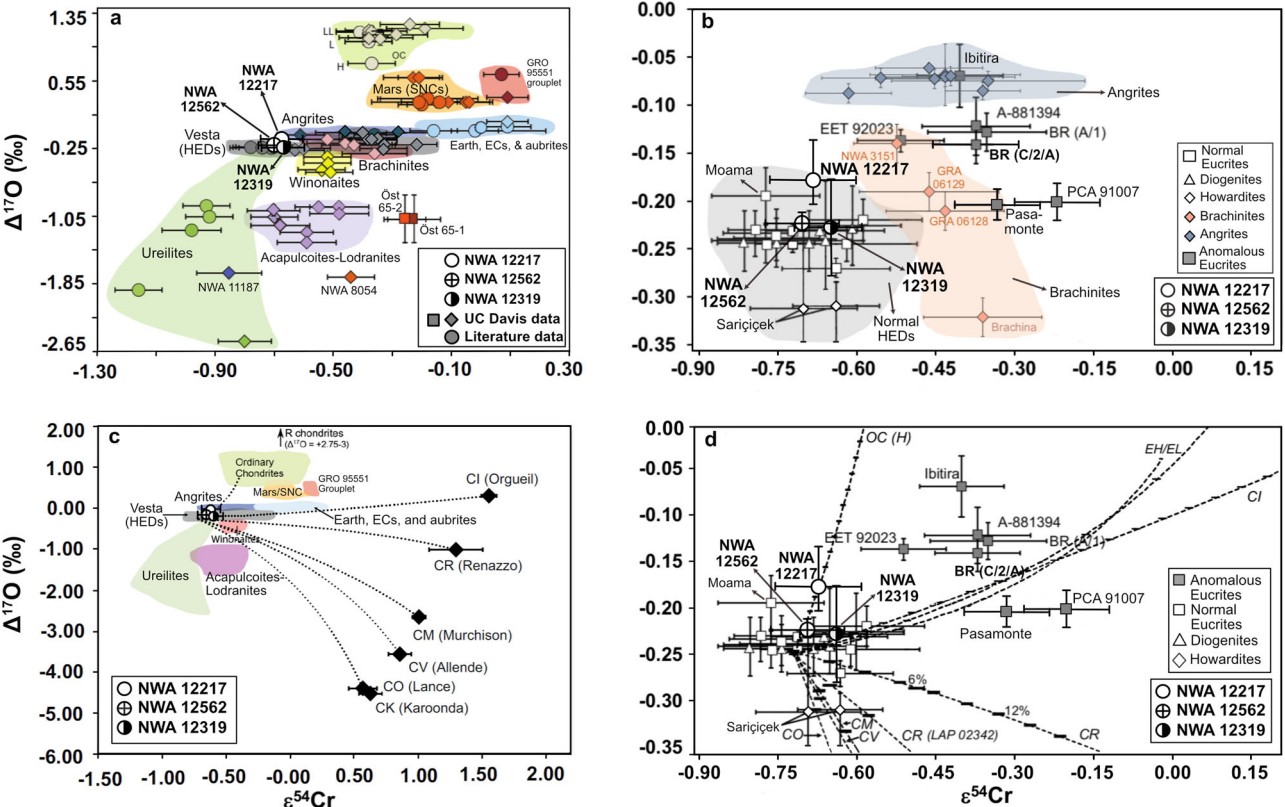

**Fig. 2 Chromium isotopic data. a** $\Delta^{17}O$ versus $\varepsilon^{54}Cr$ isotope systematics of NWA 12217, 12319, and 12562 (graphic circles) shown along with non-carbonaceous chondrites and achondrites. Colored-filled circles represent literature data while the diamonds and squares are data measured at UC Davis. The fields for ordinary chondrites (OC), Mars (SNCs), Earth and earth-likes, Vesta (HEDs), brachinites, ureilites, winonaites (win), acapulcoites (acp)/ lodranites/ungrouped achondrites (ung) are marked with select representative samples with available data. Symbol colors indicate meteorite type or grouping. Note that the carbonaceous chondrites and affiliated achondrites plot outside the field of this plot with highly positive $\varepsilon^{54}Cr$ (shown in panel **c**). **b** $\Delta^{17}O$-$\varepsilon^{54}Cr$ diagram showing NWA 12217, 12319, and 12562 (graphic circles) in comparison with the brachinites, Brachina, GRA 06128, GRA 06129, and NWA 3151 (pink diamonds), angrites, along with the normal HEDs (open symbols) and anomalous eucrites (gray filled squares) (BR = Bunburra Rockhole). The y-axis error bars for NWA 12217 and 12562 represent the range of $\Delta^{17}O$ measured, and for NWA 12319 it is the analytical error of a single measurement. **c** Overview $\Delta^{17}O$-$\varepsilon^{54}Cr$ plot showing the three new ultramafic achondrites with relation to major fields of planetary materials in non-carbonaceous (NC) and carbonaceous chondrites (CC) and mixing lines between normal HEDs with ordinary, enstatite, and carbonaceous chondrites. **d** $\Delta^{17}O$-$\varepsilon^{54}Cr$ plot of NWA 12217, 12319, and 12562 (graphic circles) along with the normal HEDs (open symbols) and anomalous eucrites (gray filled squares). Open squares represent literature normal eucrite data (BR = Bunburra Rockhole; A = Asuka). Literature data are from refs. [39,62,63,65–70] and references therein. Various dashed lines represent mixing curves between the average eucrite end member and specific chondritic end-members as indicated in panel **c** or **d**.

than twice that of the new ultramafic achondrites. Its composition is typical of plutonic angrites and angrites in general, which have superchondritic Ca/Al and higher amounts of $Cr_2O_3$ in their olivines than these new meteorites. Also, these ultramafic meteorites are clearly separated from the angrite field in $\Delta^{17}O$-$\varepsilon^{54}Cr$ systematics (Fig. 2b). It is therefore unlikely that NWA 12217, 12319, or 12562 bear affinities to the angrite parent body.

The brachinites and BLA are the most obvious analogs to the ultramafic achondrites because they are dunites and peridotites. However, they are geochemically distinct in that their range in forsterite content and trace element abundances is much more variable. Olivine CaO and $Cr_2O_3$ content of brachinites is more variable and in general higher than in the ultramafic achondrites (Fig. S8). As shown from $\Delta^{17}O$-$\varepsilon^{54}Cr$ systematics, brachinites and BLA are distinct from the three ultramafic achondrites (Fig. 2b), suggesting the two lithologies are unlikely to be of common origin. The ITE and HSE abundances of brachinites are also highly variable, likely due to their origin as residues of partial melting on a parent body that never fully differentiated[17,37,45,46]. The HSE abundances of the ultramafic achondrites plot well

below those of the brachinites and BLA, effectively ruling out association with those meteorites and any other primitive achondrites that form as residues of low degrees of partial melting. This distinction is also evident when comparing Fe/Mn versus Fe/Mg (Fig. 4). In a reducing environment, a residue of partial melting produces a linear trend with a near-constant, chondritic Mn/Mg ratio, while a cumulate produced by fractional crystallization results in a range of Fe/Mg with constant Fe/Mn[37,47]. The ultramafic achondrites show a constant Fe/Mn ratio and a range of Fe/Mg, as would be expected for cumulate rocks, while many of the primitive achondrites, including brachinites and BLA, show the constant Mn/Mg ratios expected for residues of partial melting.

The ultramafic achondrites are most similar geochemically to diogenites and the Mg-rich harzburgites found as clasts in howardites[48]. Low olivine CaO and $Cr_2O_3$ contents and the low variance in analyses from all three rocks are similar to those reported for the HED clan[15] (Fig. S7). Olivine from the HED meteorites and Mg-rich harzburgites, like the ultramafic achondrites, show a constant Fe/Mn ratio and variable Fe/Mg. In particular, the Mg-rich harzburgites overlap with NWA 12562

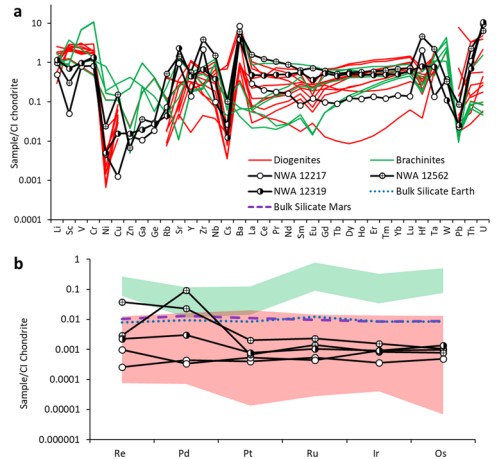

**Fig. 3 Trace element and highly siderophile element geochemistry.** CI-chondrite normalized[71] **a** ITE abundances for NWA 12217 (open circles), NWA 12319 (half-filled circles), NWA 12562 (crossed circles), brachinites[17,72] (green), and diogenites[15] (red). **b** Bulk CI-normalized HSE abundances of ultramafic achondrites plotted with those of brachinites[17] diogenites[73] and estimates for bulk silicate Earth and Mars[41].

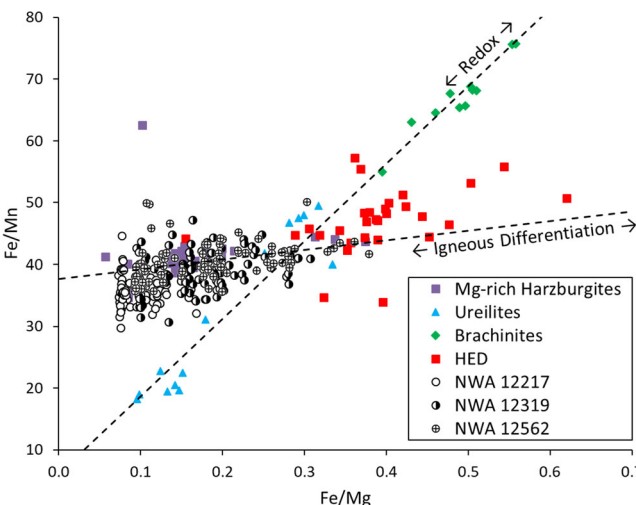

**Fig. 4 Plot of Fe/Mg versus Fe/Mn of olivine in the ultramafic achondrites and other meteorite groups[15,47,48,72,74,75].** Mg-rich harzburgites are ultramafic clasts found within howardites. HED olivine analyses are overwhelmingly from diogenites. Primitive achondrites such as the ureilites and brachinites plot with constant Mn/Mg, while NWA 12217, 12319, and 12562 plot with relatively constant Fe/Mn and variable Fe/Mg.

and 12319, with the latter showing significantly more spread in Fe/Mg. Because the ultramafic achondrites are breccias, this greater variability is likely due to impact sampling and mixing of fragments from a large differentiated protolith.

The similar shape of the REE patterns of NWA 12217 and 12562 suggests that they were formed by similar processes, while the bulk abundance differences are likely an artifact of brecciation processes that led to variance in modal pyroxene abundance. Residues of partial melting are expected to have LREE/HREE < 1[49]. Instead, NWA 12217 and 12562 show enrichments in the LREE relative to the HREE, suggesting different formation conditions such as accumulation in a fractionating magmatic assemblage within a region of magma storage in their parent body. Although LREE enrichment is also possible through incorporation of crustal material during brecciation or

metasomatism, such crustal incorporation is unlikely because both meteorites appear to be monomict breccias with no evidence of contamination by a different lithology. NWA 12319 contains three times as much $P_2O_5$ as the other two rocks, so it is possible that its flat REE pattern is a result of more contributions to its REE budget from phosphate, or potentially a trapped melt component, relative to pyroxene.

Importantly, the HED meteorites show up to five orders of magnitude of variations in HSE concentrations[41], so this metric alone is not sufficient to establish a relationship of the ultramafic achondrites to the diogenites. However, in combination with other lines of evidence discussed above, it rules out association with other olivine-rich planetary material. NWA 12217, 12319, and 12562 have broadly chondritic HSE patterns that are found in all planetary bodies that experienced core-mantle differentiation[41]. This implies that they originate from bodies that experienced core formation events and whose mantles were later enriched in the HSE from late accretion of chondritic material. Their similarity to diogenites and Mg-rich harzburgites suggests that these parent bodies are sampled by V-type asteroids.

The Fe/Mn systematics of the olivines in NWA 12217, 12319, and 12562 argue convincingly for a cumulate origin within a fractionating magma chamber for these meteorites. The diogenites and Mg-rich harzburgites included in howardites are similarly cumulate rocks within this scheme, but whether they originated on the same parent body is not fully resolved. The pyroxene minor element diversity (Fig. S9) exhibited in particular by NWA 12319 and 12562 overlaps and broadens the fields plotted by the diogenites, Mg-rich harzburgites, and NWA 12217. Coupled with the oxygen and chromium isotopic evidence, this suggests that the ultramafic achondrites could have originated on Vesta or a V-type asteroid.

The Dawn mission arguably created more questions regarding Vesta and the HED meteorites than it solved by observing a large metallic core over 100 km in diameter[50] and not detecting large volumes of olivine-rich mantle material exposed by deep impact craters[51]. If the bulk composition of Vesta is chondritic, the elevated REE contents of the crustal eucrites argue for the existence of a complementary REE-depleted and olivine-rich mantle seven to ten times the size of the crust[51,52]. However, the large Vestan core and the deep crust-mantle boundary suggested by impact craters that excavated olivine-free material from 100 km deep[53], severely limit the potential size of this mantle. The discovery of the ultramafic achondrites is in line with these observations, as they do not appear to represent a large uniform mantle that crystallized from a magma ocean.

The petrological and geochemical characteristics of the ultramafic achondrites favor serial magmatism within fractionating magma chambers on their parent bodies, just as the discovery of olivine-rich material overlying eucritic material favors this model on Vesta[54]. The variability in Mg# and pyroxene compositions found within all three breccias suggests that they are sampling heterogeneous igneous bodies, whereas under a magma ocean scenario more homogeneity should be expected. The REE compositions of the ultramafic achondrites are likewise varied and dependent on high-Ca pyroxene content, a phase that is not expected to be in the mantle of a sodium-depleted parent body such as Vesta[51]. While bulk analyses of the ultramafic achondrites suggest they are as alkali-depleted as the HED meteorites, brecciated ultramafic rocks are unlikely to precisely represent the volatile budgets of their host bodies (Fig. S10). Thus, the compositions of these meteorites raise additional complexities in piecing together the igneous histories of Vesta, the vestoids, and any other differentiated planetesimals that produced fractionating magmas.

## Methods

**Sample preparation.** Several fragments from deposit samples of NWA 12217 and 12562 were mounted in epoxy and polished to 0.05 μm smoothness. Two polished thin sections were also produced from fragments of NWA 12217 and 12319 (BC2920.2). These samples were analyzed by optical microscopy, electron probe microanalysis (EPMA), and scanning electron microscopy (SEM) at the Institute of Meteoritics, UNM, Albuquerque, NM (NWA 12217, 12319, and 12562) and the *Institut für Geowissenschaften*, Christian-Albrechts-Universität in Kiel, Germany (NWA 12319).

**Oxygen isotopic analysis.** Six fresh fragments of interior materials of NWA 12217 and NWA 12562, weighing between 1 and 2 mg, were selected using a stereomicroscope to avoid any possible contamination from fusion crust or terrestrial staining. Oxygen isotopic analyses were performed using a $CO_2$ laser + $BrF_5$ fluorination system following modified procedures outlined in Sharp[55]. The analyses in Göttingen were performed following the protocol described in Pack et al.[56] and Peters et al.[57], and details regarding the UNM technique are given in Wostbrock et al.[58]. Oxygen isotope compositions were calculated using the following procedure: the $\delta^{17,18}O$ values refer to the per mil deviation of a sample's $(^{17}O/^{16}O)$ and $(^{18}O/^{16}O)$ ratios from the V-SMOW standard values, respectively, expressed as $\delta^{17,18}O = [(^{17,18}O/^{16}O)_{sample}/(^{17,18}O/^{16}O)_{V-SMOW} - 1] \times 10^3$. The delta values were then converted to logarithmic $\delta'$ in which $\delta^{17,18}O' = \ln(\delta^{17,18}O/1000 + 1) \times 1000$. The $\Delta^{17}O'$ values were obtained from the $\delta'$ values following $\Delta^{17}O' = \delta^{17}O' - 0.528 * \delta^{18}O'$. Typical analytical precision of the laser-fluorination technique is better than ±0.02‰ for $\Delta^{17}O'$.

**Trace element and highly siderophile element analysis.** Whole-rock fragments of all three meteorites were partially disaggregated to obtain small chips that were subsequently crushed with limited force in an agate mortar and pestle to make a partially representative (~0.3 g) whole-rock powder for each sub-fragment. Bulk rock major- and trace-element abundances were determined at the *Scripps Isotope Geochemistry Laboratory (SIGL)* on sample powders by digestion in sealed Teflon vessels at 140 °C in Optima grade concentrated HF and $HNO_3$ (4:1 V/V) for >72 h on a hot plate, along with total analytical blanks and terrestrial basalt standards. Samples were sequentially dried and taken up in concentrated $HNO_3$ to remove fluorides, followed by dilution and doping with indium to monitor instrumental drift during analysis. Major- and trace-element abundance analyses were obtained using a *ThermoScientific* iCAP Qc quadrupole inductively coupled plasma mass spectrometer in normal mode. Reference materials were analyzed as "unknowns" (BIR-1, BHVO-2, HARZ-01, and BCR-2) to assess external reproducibility and accuracy (see also Day et al.[46]). For major and trace elements, reproducibility of reference materials was better than 5% (RSD).

Osmium isotope and HSE abundance analyses were performed at the SIGL using methods described in Day et al.[41]. Homogenized powder aliquots of the samples and total procedural blanks were digested in sealed borosilicate Carius tubes, with isotopically enriched multi-element spikes ($^{99}Ru$, $^{106}Pd$, $^{185}Re$, $^{190}Os$, $^{191}Ir$, $^{194}Pt$), and 6 mL of a 1:2 mixture of multiply Teflon distilled HCl and $HNO_3$ that was treated with $H_2O_2$ to expunge Os. Samples were digested to a maximum temperature of 270 °C in an oven for 72 h. Osmium was triply extracted from the acid using $CCl_4$ and then back-extracted into HBr, prior to purification by micro-distillation that was performed twice. Rhenium and the other HSE were recovered and purified from the residual solutions using standard anion exchange separation techniques. Isotopic compositions of Os were measured in negative-ion mode on a *ThermoScientific* Triton thermal ionization mass spectrometer at the SIGL. Rhenium, Pd, Pt, Ru, and Ir were measured using a *Cetac* Aridus II desolvating nebulizer coupled to a *ThermoScientific* iCAP q ICP-MS. Offline corrections for Os involved an oxide correction, an iterative fractionation correction using $^{192}Os/^{188}Os = 3.08271$, a $^{190}Os$ spike subtraction, and finally, an Os blank subtraction. Precision for $^{187}Os/^{188}Os$, determined by repeated measurement of the UMCP Johnson-Matthey standard was better than ±0.2% (2 St. Dev.; 0.11390 ± 20; $n = 5$). Measured Re, Pd, Pt, Ru, and Ir isotopic ratios for sample solutions were corrected for mass fractionation using the deviation of the standard average run of the day over the natural ratio for the element. External reproducibility on HSE analyses was better than 0.5% for 0.5 ppb solutions and all reported values are blank corrected. The total analytical blank run with samples had $^{187}Os/^{188}Os = 0.177 ± 0.02$, with quantities (in picograms) of 0.9 [Re], 1.9 [Pd], 1.4 [Pt], 2.1 [Ru], 1.2 [Ir], and 0.3 [Os]. Blank contributions represent <10% in NWA 12562, but >40% for Re, Pd, Pt, and Ru for NWA 12217. We measured NWA 12217 and NWA 12562 in duplicate and NWA 12319 once during the course of this study.

**Chromium isotopic analysis.** An interior fusion-crust-free chip of each sample was ground to a powder using an agate mortar and pestle. For dissolution, an aliquot (~15 mg) of the whole-rock powder was placed into a PTFE capsule with a 3:1 mixture of concentrated $HF:HNO_3$ that was sealed in a stainless-steel Parr bomb jacket. The Parr bomb was heated at 190 °C for 96 hours in an oven. After complete dissolution, the sample was processed through a 3-column chemistry extraction procedure to separate Cr from all other matrix elements[59]. Cr recovery yields for the three ultramafic samples are all above 99% for the first column, 92–94% for the second column, and 91–97% for the third column, with a total cumulative yield of 84–88%. The purified Cr fractions were loaded onto outgassed

W filaments (the total Cr load split among four filaments) and each set of sample filaments bracketed with two filaments loaded with a similar amount of terrestrial Cr standard (NIST SRM 979) before and after. The Cr isotopic composition was measured at UC Davis using a Thermo Triton Plus thermal ionization mass spectrometer[60]. Each filament analysis consisted of 1200 ratios with 8 s integrations times. A gain calibration was made at the beginning of each filament, the baseline was measured, and the amplifiers were rotated between each block of 25 ratios. The mass fractionation was corrected using an exponential law and a $^{50}Cr/^{52}Cr$ ratio of 0.051859[61]. The $^{53}Cr/^{52}Cr$ and $^{54}Cr/^{52}Cr$ ratios are expressed in ε-notation (that is, parts per 10,000 deviation from the NIST SRM 979 Cr standard).

## Data availability

The authors declare that all data supporting the findings of this study are available within the paper and its Supplementary Information files.

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

## Author contributions

Sample preparation, optical petrography, and scanning electron microscopy were performed by Zoltan Vaci. Electron microprobe data were collected by Zoltan Vaci and Rainer Bartoschewitz. Oxygen isotopic analysis was performed by Karen Ziegler and Andreas Pack. Trace element and highly siderophile element geochemistry were analyzed by James Day and Marine Paquet. Chromium isotopic analysis was performed by Qing-zhu Yin, Audrey Miller, and Supratim Dey. Initial characterization of meteorites was performed by Carl Agee.

## Competing interests

The authors declare no competing interests.
