## [Peer Review File · Nature Communications]

REVIEWER COMMENTS

Reviewer #1 (Remarks to the Author):

The manuscript "End of the Solar System asteroidal dunite shortage" presents the discovery and characterization of three olivine-dominated meteoritic samples, whose compositional characteristics suggest they come from the mantle of a differentiated body as well as a possible connection with the HED family of meteorites associated with asteroid Vesta.

While we possess basaltic and metallic meteorites that represents fragments of the basaltic crusts and the metal cores of early-differentiated asteroids, the olivine-dominated meteorites that should represent the mantles of such differentiated-the-destroyed bodies are notably lacking from the terrestrial collection. The lack of these mantle fragments has been an outstanding open question in the fields of meteoritic, asteroidal and planetary formation studies for at least the past quarter of century.

As a result, the discovery of these three meteorites introduces a new dimension in the study the differentiation processes that shaped the earliest generation of planetary bodies in the Solar System and, more generally, the planet formation processes that formed them in the solar nebula.

The manuscript is quite interesting, clearly written and offers a detailed discussion of the main compositional characteristics of the three olivine-dominated meteorites. The main thread of my comments below is that the manuscript could benefit from a more straightforward discussion of the implications of these meteorites for both their putative parent body Vesta and the early history of the Solar System.

Specifically, if these olivine-dominated meteorites come from Vesta, it should be clarified how they fit in the context of the view provided by the HEDs and Dawn. On the other hand, taking the data of Fig. 1B and 2A-B at face value, there always seem to be one of the three meteorites not fitting in the pattern of the other two. Some of these discrepancies are discussed in the text, but I admit that at the end of the manuscript I was left with the doubt of whether the three meteorites could have more than one parent body.

Some of the questions asked in the following comments may be even naive for experts, but even in this case (possibly more, actually) addressing them would be valuable to the wider audience of the article.

Major comments

Composition of the olivine-rich meteorites and Vesta

The text presents a discussion of different compositional features of the three meteorites in terms of Mg#, oxygen isotopic composition, REE & HSE enrichments. While the data are discussed in details and the answers to the following questions are probably straightforward to experts, it is not immediately apparent how well the characteristics of these olivine-dominated meteorites fit in the global picture of Vesta's composition. Specifically, Consolmagno et al. (2015) pointed out the open issues on Vesta's physical and compositional nature left unsolved, and sometimes actually made worse, by the comparison of Vesta's, HED's and Dawn's data. Some aspects linked to these issues that could benefit from brief discussion or clarification are:

- REEs and ITEs: the enrichment of REE in eucrites has been interpreted as suggestive of Vesta's crust representing 10-15% of Vesta's mass (Consolmagno & Drake 1977, Consolmagno et al. 2015). Crustal thickness is one of the unknown parameters that act as a constraint in modelling the interior from gravity data (e.g. Russell et al. 2012, Ermakov et al. 2014) and is an important constraint for modelling the collisional environment of the early Solar System (Turrini et al. 2018 and references therein). Does the abundance of REE of the olivine-dominated meteorites fits or deviates from this picture?

Also, I was noticing in Fig. 1A that NWA 12319 has a distinctly different REE enrichment pattern with

respect to the other samples (which, as pointed out by the authors at lines 179-186, are instead quite similar) and, differently from the other two samples, has a negative Eu anomaly. Possibly these differences are not significant, but this is not mentioned in the relevant discussion, which is focused on the similarities between NWA 12217 and 12562.

- Mg#: Consolmagno et al. (2015) and previous authors cited therein made a case for the Mg# of Vesta's mantle being no higher than 0.8. NWA 12319 and 12562 fit this constraint, while NWA 12217 is a bit higher (0.82). What is the impact of this difference? Is it negligible or does it require changes to the models, e.g. in the estimated metallic core radius?
Connected to this: is there a direct correlation between the Mg# and the Fe/Mg ratio shown in Fig. 3? Would it make sense to have the Mg# on the upper axis?

- HSE enrichment (specifically lines 193-199): the authors make a point about the HSE enrichment and the late accretion of chondritic material. Is this material supposed to be accreted initially in the crust and to sediment in the mantle later, or to be deposited directly in the mantle?
In the second case, given the estimated crustal thickness and the scaling of the penetration depth of the impactors with their diameters, a direct delivery to the mantle would imply the impact of projectiles larger than at least 10 km, and possibly of the order of 40 km. This size range is similar to that of the impactors invoked to create the basins of Rheasilvia and Veneneia at the South Pole of Vesta (Ivanov & Melosh 2013; Jutzi et al. 2013), so it would have major implications for the survival of the crust (see Turrini et al. 2018 and references therein).
If instead these HSEs are derived from material deposited initially in the crust by impacts (i.e. the mantle acts as a sink), would this change the upper limit to the accreted chondritic material estimated in previous work by one of the authors based on the values of diogenites (Day et al. 2012, ~1% in mass of the total mass of Vesta). This value provides an important constraint on the primordial collisional environment of the asteroid belt, so clarifying the impact of the olivine-dominated meteorites on it can have far reaching implications beyond the specific study of Vesta.

- Oxygen isotopic abundances (specifically lines 143-145 plus subsequent discussion): in the manuscript it is stated that the oxygen isotopic composition of the three olivine-dominated meteorites suggests affinities to other families of meteorites like brachinites and angrites. This seems a bit at odds with what is shown in Fig. 1B, where NWA 12319 and 12562 have values at the boundary between the high-end values of the oxygen-similar eucrites and the lower boundaries to the d17O values of oxygen-anomalous eucrites. NWA12217 falls instead between the angrites and the oxygen-anomalous eucrites (Scott et al. 2009; Zhang et al. 2019 and references within). I may be misunderstanding the plotted data but, given the general interpretation of the oxygen-anomalous HEDs and taking Fig. 1B at face value, it would seem that the olivine-dominated meteorites may come from at least two different bodies.

Other comments

- Line 1 (Title), lines 25-27 (last sentence of the abstract/introductory paragraph), line 75: the manuscript links the discovery of the three olivine-dominated meteorites to the end of the "great dunite shortage", i.e. the lack/paucity of olivine-dominated meteorites/asteroids.
The discovery of the three meteorites is a step in the direction of ending this open issue but stating that it marks its end is too definitive a statement, since we still have only a limited number of samples that the authors suggest being linked to only one body.
If the samples are all related to Vesta, they do mark an end to the lack of vestan mantle samples (another major open issue, as testified by the number of thematic studies linked to the Dawn mission), but in this case these sentences should be reframed in the specific vestan context.

- Lines 73-74: the reference to the Nice model is incorrect here: while early claims by the original series of papers argued for major effects on the asteroid belt, particularly in terms of excitation and ejections, later works (Minton & Malhotra 2009; Morbidelli et al. 2010; Pirani et al. 2016, with the

latter also exploring the collisional implications) have clearly shown that the effects of the Nice Model on the asteroid belt are much more limited than originally claimed (basically, less than the combined effects of the subsequent 0.1-1 Gyr).

In terms of the Grand Tack: it can provide an explanation but its complex migration pattern is hardly required. Turrini et al. (2012) have shown that the formation of Jupiter is condition enough (migration is important but is not strictly required) for the collisional erosion of early formed planetesimals. Later works by Raymond & Izidoro (2017) and Pirani et al. (2019) confirmed, with more detailed disk models (and in the case of the latter more self-consistent migration scenarios), the natural emergence of the conditions to trigger the collisional erosion described by Turrini et al. (2012) without the need of any ad-hoc condition (like in the Grand Tack).

- Lines 211-221: there are additional sites where olivine has been suggested by different authors (Ruesch et al. 2014; Nauthes et al. 2015; Palomba et al. 2015; see Turrini et al. 2016 for a global map), all associated with shallow craters. At the same time, the presence of olivine has been ruled out inside the two South Polar basins and on the central mount of Rheasilvia (Ammannito et al. 2013; Ruesch et al. 2014), which seems to indicate the lack of widespread olivine plutonic deposits over a range of depths and an area of the order of 25% the surface of Vesta.

This dichotomy is what brought Consolmagno et al. (2015) to point out the difficulties in quantitatively constraining the internal structure of Vesta and, tangentially, subsequent authors to suggest that the olivine detected on Vesta could have a exogenous origin (Nathues et al. 2015; Le Corre et al. 2015; Turrini et al. 2016).

Can the serial magmatism in large magma chambers be reconciled with this dichotomy in the olivine distribution? Do the petrological and geochemical characteristics of the three meteorites provide additional insight on the issue of the distribution of olivine on Vesta?

Minor comments:

- Lines 34-36: one could make the point that the existence of iron meteorites belonging to tens of compositional groups, notwithstanding the depletion of the asteroid belt, observationally supports the theoretical argument.

- this is mostly a matter of style, but in various point in the manuscript the adjective "Vestal" is used to mean "of Vesta". At the time of the arrival of Dawn to Vesta there has been quite a debate on the correct adjective to use, as "Vestal" has a very specific connotation (connected to the religious cult of the roman goddess Vesta) and was therefore not adopted. The adjective "Vestan" was considered more correct and adopted, analogously to why we use "Martian" instead of "Martial" for Mars.

Reviewer #2 (Remarks to the Author):

Review of "End of the Solar System Asteroidal Dunite Shortage," by Z. Vaci et al.

The authors have presented an interesting work regarding remarkable olivine-rich achondrites, presenting new data regarding the petrography, bulk geochemistry, mineral chemistry, oxygen isotopes, rare earth elements, and highly siderophile elements. The authors make the case that the meteorites have affinity with HED meteorites, and may thus represent a portion of the mantle of Vesta (with the caveat that they could originate from another differentiated parent body).

Overall, I find the manuscript to be well written, although some phrasing and grammatical changes are recommended below.

My biggest concern is with trying to parse the logic linking the title of the manuscript with the conclusions of the text. It may be that I have simply misunderstood an argument (in which case, I would recommend clarifying the text), but I am having trouble with the logic of the manuscript:

- 1) The title and final sentence of the abstract claim that these three meteorites represent an end to the dunite shortage of the Solar System.
- 2) The shortage of olivine-rich material is introduced as being based on the lack of observations of olivine rich asteroids or meteorites (although notably, there is a non-zero number of both).
- 3) Three new meteorites are presented and are shown to be similar to HED's, and especially Diogenites (although olivine rich instead of pyroxene rich), and other known possible parent bodies are ruled out.
- 4) The authors conclude with two possibilities:
A: The meteorites originate from an otherwise unknown object that was destroyed, thus favoring the idea that the missing mantle problem is solved via widespread destruction of such material ("Battered to bits" hypothesis).
B: The meteorites originate from Vesta, and most probably from Arruntia or Bellicia craters, where olivine rich material was excavated.

The conclusions follow from the analysis of the samples, but I am struggling to understand how either scenario A or B marks an "end" to the dunite shortage. In the case of A, it is basically a claim that an already proposed hypothesis is correct (although the Battered to bits hypothesis would not seem to be a unique explanation for these olivine-rich meteorites). In the case of B, it is a Vesta-specific claim that does not address the broader problem - this essentially argues that the deep mantle material of Vesta was never really ejected, but does not say anything about other planetesimals. Perhaps the authors are trying to claim that the plutonic origin of the meteorites, rather than a magma ocean origin, is responsible for much greater amounts of pyroxene rather than olivine crystallization? But in this case, it should be made more explicit.

I would like the authors to clarify the link between the title and the conclusions of the manuscript. This might also be aided with a summary diagram that explains the claims of the authors.

Detailed comments:

Title: As above, the "end" of the dunite shortage should be resolved, and as this study does not involve asteroids at all, "Asteroidal" should probably be removed from the title to avoid confusion.

Line 14-15: "The great dunite shortage...represent the paucity of olivine rich asteroids" is not correct; please reword.

Line 22: "Consistent with being related to" - please rephrase.

Line 23: The Cr-rich veins and symplectites are only really discussed in the supplemental information and not so much in the main text (barring a brief mention in lines 88-90). If this is a key line of evidence, please discuss it more in the main text. Especially so in context of exogenous fluid and melt interaction (which is described well in the supplement). If this is not critical, then it probably doesn't need to be in the abstract.

Line 25-27: As mentioned above, this is an important and substantial claim, but the text does not seem to quite arrive at this conclusion.

Line 32: "Either" typically refers to precisely 2 items; please rephrase.

Line 35: The hyphens around "and meteorite samples of them" should be replaced with commas or parenthesis.

Line 56: "...and complementarily there have not been..." Please rephrase.

Line 67-74: This simple explanation for the lack of olivine-rich meteorites seems fairly satisfactory. If widespread differentiation never took place, then there is not so much of a problem to be solved. Could you please clarify why the dunite shortage remains a problem?

Line 102: I am uncertain what it means for a trace phase to be "well-distributed"? Please clarify. Also, is it correct that feldspar is a trace phase, is evenly(?) distributed, is present as brecciated fragments, and is associated with pure silica?

Line 105: Feldspar as inclusions in olivine with radiating fractures might be an interesting image. Is this suggestive of a fluid inclusion?

Line 113-114: See comments on Figure 1.

Line 114: A reference to Figure 1B should be here.

Line 122: "(72)" was left in the PDF; please remove.

Line 123: "Incompatible" should be lower case.

Line 138-139: Please include a reference for how Pd can be added via terrestrial weathering.

Line 140: "Subtracting the additions..." Please clarify: does this mean "Not including" or does this mean a mathematical adjustment to the data?

Line 145-149: The argument being made here appears to be that Angrites cannot be the source of the meteorites in this study due to oxygen fugacity; however, the meteorites are also thought to represent the deep mantle of their parent body, which would likely be more reducing than the shallower portions of the parent body. It is not immediately clear to me that the oxidation state of the meteorites in this case can distinguish the parent body. Can this be clarified?

Line 156 and elsewhere: Please use "because" instead of "since."

Line 171-178: The comparison with diogenites and Howardites does not mention oxygen fugacity, which was a primary listed reason for rejecting Angrites.

Line 189: "the incorporation" is difficult to parse; please change the phrasing.

Line 187-192: This section can be made more compact by combining sentences and rephrasing. I am not certain that it is critical in any case, and this discussion of potential LREE enrichment could perhaps be expanded and migrated to the supplementary information; it seems like it would fit well with the symplectite discussion.

Lines 194 and 199: The statement is made that HSEs are not sufficient for establishing a relationship with diogenites, but later in the paragraph, the similarity of the HSEs to diogenite patterns are said to be suggestive of a relationship with Vesta. Please rephrase this paragraph to resolve this apparent contradiction.

Line 208: In the paragraph explaining possible explanations for the dunite shortage (lines 54-66), the "Battered to Bits" hypothesis is not mentioned. Please discuss this earlier in the text.

Line 216: "...the study argues..." should be "...this study argues..."

Line 217-219: This is interesting; can the relevance to the HEDs of plutonism as opposed to magma ocean be discussed?

Line 220-221: I take this to mean that the reason we don't have more dunitic material from Vesta is that it was never excavated and ejected?

Line 222: A paragraph explaining how the dunite shortage is solved would be appreciated here.

Line 225: How many thin sections were made?

Figure 1: Please label the phases in A. Images of the other two samples would also be appreciated, even if they are in the supplemental information. In B, it appears that you have plotted Iberita and A-881394 O isotopes. As eucrites, these samples are a bit unusual in their 17-O isotopic composition. Perhaps they should be labeled as outliers. Also, this study argues that the meteorites are most closely related specifically to diogenites and a subset of Howardites; it would be good to plot the H, E, and D samples with distinct symbols. Further, the samples of NWA 12217 appear to have significantly higher 17-O isotopic composition than the other samples. I cannot find a data table with the O isotopes in the submitted files to try to replot these data. This graph does not look consistent with the statement that the samples are indistinguishable. Is it correct to assume that the error bars associated with these data points are smaller than the symbols?

Figure 2: Please include a legend on this figure. The chondrite normalization is Anders and Grevesse, 1989; was there a reason this was used instead of a more recent reference, such as Lodders et al., 2009? B: I am accustomed to seeing the HSEs in order of vaporization; is there a reason for the order presented here?

Figure 3: Are the Mg-rich Harzburgites the same as the Howardites? Please make this clear. As before, it would make sense to separate Howardite, eucrite, and diogenite samples in this plot, as it is relevant to the details of the interpretation.

Supplementary information
Text appears to be fine.

It would be nice if additional thin section images could be shown here to support the textural interpretations made in the main text.

Figure S2: Sample NWA 12217 appears in this diagram, and also in several others, to plot far away from the other two. The ^{17}O isotopic composition is higher, the Mg# of olivine is higher, the modal percentage of olivine is higher, and the feldspars are quite distinct. It also has lower Sc, Cu, and REE abundances than the other two samples. Could you please include a summary of why the three samples all originate from the same parent body?

Figure S6-S7: These images are beautiful.

Reviewer #3 (Remarks to the Author):

Dear Dr. Vaci et al.,

Here you will find my review of the manuscript "End of the Solar System asteroidal dunite shortage".

Identifying and characterizing meteoritic material that may represent the mantle of differentiated asteroids is an important endeavor that will provide insights into one of the earliest stages of asteroid differentiation. The manuscript shows that 'missing' dunites have been identified and reports their characterization in detail. While that itself is interesting, the manuscript convincingly argues that the studied samples (NWA 12217, 12319, and 12562) may be from asteroid Vesta, adding significant weight to the importance of this work.

The manuscript is well-written, well-presented, and was a pleasure to read. I thought it laid out the so-called 'dunite shortage' issue well and made a convincing argument that such samples have now been found. I believe that this work will be of substantial interest to the extraterrestrial sample science community. I have no hesitation recommending this work for publication in Nature Communications after minor revisions.

Suggestions that should be addressed, include:

1. Line 14: Introduce what a dunite is for the non-expert reader.
2. Line 118/Fig. 1B: These samples do not seem consistent with brachinites. HEDs/angrites? Sure. But they do not plot with the brachinites. I suggest cutting mention of brachinites here.
3. Line 122: Remove struck through parentheses.
4. Fig. 2: Add legend for symbols to graph; this is easier for the reader than having to dig through the caption.
5. Line 144: Once again, the O-isotopes of these samples do not seem consistent with brachinites. This supports your later conclusion that these samples are not related to brachinites.
6. Supplementary data set: some of the microprobe data are of poor quality (totals <98 wt.%) and should not be reported. It appears that many (but not all) of these analyses have been flagged as poor quality in the annex but I suggest removing them from the data file and all relevant figures. They should also be removed from the calculations of Mg# if they were included. Furthermore, cations are not reported for any of these analyses; stoichiometry of all analyses should be calculated to check the quality of the reported data prior to publication.

I look forward to seeing this manuscript in print.

REVIEWER COMMENTS

(Authors' replies in red)

Reviewer #1 (Remarks to the Author):

The manuscript “End of the Solar System asteroidal dunite shortage” presents the discovery and characterization of three olivine-dominated meteoritic samples, whose compositional characteristics suggest they come from the mantle of a differentiated body as well as a possible connection with the HED family of meteorites associated with asteroid Vesta.

While we possess basaltic and metallic meteorites that represents fragments of the basaltic crusts and the metal cores of early-differentiated asteroids, the olivine-dominated meteorites that should represent the mantles of such differentiated-the-destroyed bodies are notably lacking from the terrestrial collection. The lack of these mantle fragments has been an outstanding open question in the fields of meteoritic, asteroidal and planetary formation studies for at least the past quarter of century. As a result, the discovery of these three meteorites introduces a new dimension in the study the differentiation processes that shaped the earliest generation of planetary bodies in the Solar System and, more generally, the planet formation processes that formed them in the solar nebula.

The manuscript is quite interesting, clearly written and offers a detailed discussion of the main compositional characteristics of the three olivine-dominated meteorites. The main thread of my comments below is that the manuscript could benefit from a more straightforward discussion of the implications of these meteorites for both their putative parent body Vesta and the early history of the Solar System. Specifically, if these olivine-dominated meteorites come from Vesta, it should be clarified how they fit in the context of the view provided by the HEDs and Dawn. On the other hand, taking the data of Fig. 1B and 2A-B at face value, there always seem to be one of the three meteorites not fitting in the pattern of the other two. Some of these discrepancies are discussed in the text, but I admit that at the end of the manuscript I was left with the doubt of whether the three meteorites could have more than one parent body. Some of the questions asked in the following comments may be even naive for experts, but even in this case (possibly more, actually) addressing them would be valuable to the wider audience of the article.

Given the new Cr isotope data added to the manuscript, our conclusions regarding parent body origin have been changed. These meteorites are not associated with the main-group isotope-normative HEDs, and rather are likely related to the Vestoids.

Major comments

Composition of the olivine-rich meteorites and Vesta

The text presents a discussion of different compositional features of the three meteorites in terms of Mg#, oxygen isotopic composition, REE & HSE enrichments.

A note on O isotopes and Mg#: Additional O isotopic analyses have been conducted since initial submission on NWA 12217 and 12562 and the results have been included. They do not change our interpretations of the original O isotope data. The provided Mg#s were incorrect in the initial submission and have been corrected. They are now expressed in molar terms [Mg/(Mg+Fe)]. New Cr isotopic data has also been included, and these do change our discussion and conclusions regarding parent body origin.

While the data are discussed in details and the answers to the following questions are probably straightforward to experts, it is not immediately apparent how well the characteristics of these olivine-dominated meteorites fit in the global picture of Vesta's composition. Specifically, Consolmagno et al. (2015) pointed out the open issues on Vesta's physical and compositional nature left unsolved, and sometimes actually made worse, by the comparison of Vesta's, HED's and Dawn's data. Some aspects linked to these issues that could benefit from brief discussion or clarification are:

Added a discussion on the Dawn mission and its confusing conclusions in light of the HEDs and these meteorites. Again, these meteorites seem to be Vesta/Vestoid-like, but not necessarily from Vesta.

- REEs and ITEs: the enrichment of REE in eucrites has been interpreted as suggestive of Vesta's crust representing 10-15% of Vesta's mass (Consolmagno & Drake 1977, Consolmagno et al. 2015). Crustal thickness is one of the unknown parameters that act as a constraint in modelling the interior from gravity data (e.g. Russell et al. 2012, Ermakov et al. 2014) and is an important constraint for modelling the collisional environment of the early Solar System (Turrini et al. 2018 and references therein). Does the abundance of REE of the olivine-dominated meteorites fits or deviates from this picture?

We appreciate this point and have thus rewritten the final 3 paragraphs of the paper to accommodate these issues. Given that we cannot now conclude that these meteorites are Vestan, there is no longer a contradiction.

Also, I was noticing in Fig. 1A that NWA 12319 has a distinctly different REE enrichment pattern with respect to the other samples (which, as pointed out by the authors at lines 179-186, are instead quite similar) and, differently from the other two samples, has a negative Eu anomaly. Possibly these differences are not significant, but this is not mentioned in the relevant discussion, which is focused on the similarities between NWA 12217 and 12562.

Added note on NWA 12319 REE pattern and the possibility of phosphates influencing the shapes and abundances. As these meteorites are all breccias, sampling bias is an issue.

- Mg#: Consolmagno et al. (2015) and previous authors cited therein made a case for the Mg# of Vesta's mantle being no higher than 0.8. NWA 12319 and 12562 fit this constraint, while NWA 12217 is a bit higher (0.82). What is the impact of this difference? Is it negligible or does it require changes to the models, e.g. in the estimated metallic core radius?

Since we are favoring a serial magmatism model, with multiple large magma chambers, the bulk Mg# constraint does not necessarily apply to a few samples of one or more of these magma chambers, which could have large degrees of Mg# variability due to fractional crystallization. Upon reading Consolmagno et al (2015), they leave relatively open the question of the Vestan mantle's Mg#, citing multiple estimates and suggesting that the larger the iron core is, the higher Mg# one should expect for the mantle.

Connected to this: is there a direct correlation between the Mg# and the Fe/Mg ratio shown in Fig. 3? Would it make sense to have the Mg# on the upper axis?

The Fe/Mg ratio is just Mg# expressed differently, and in the literature, one sees this chart with either value. The importance of this figure is the constant Fe/Mn ratio that does not change with Mg# (or Fe/Mg).

- HSE enrichment (specifically lines 193-199): the authors make a point about the HSE enrichment and the late accretion of chondritic material. Is this material supposed to be accreted initially in the crust and to sediment in the mantle later, or to be deposited directly in the mantle? In the second case, given the estimated crustal thickness and the scaling of the penetration depth of the impactors with their diameters, a direct delivery to the mantle would imply the impact of projectiles larger than at least 10 km, and possibly of the order of 40 km. This size range is similar to that of the impactors invoked to create the basins of Rheasilvia and Veneneia at the South Pole of Vesta (Ivanov & Melosh 2013; Jutzi et al. 2013), so it would have major implications for the survival of the crust (see Turrini et al. 2018 and references therein). If instead these HSEs are derived from material deposited initially in the crust by impacts (i.e. the mantle acts as a sink), would this change the upper limit to the accreted chondritic material estimated in previous work by one of the authors based on the values of diogenites (Day et al. 2012, ~1% in mass of the total mass of Vesta). This value provides an important constraint on the primordial collisional environment of the asteroid belt, so clarifying the impact of the olivine-dominated meteorites on it can have far reaching implications beyond the specific study of Vesta.

We do not know the timing or extent of HSE delivery, nor do we know the thickness of the crust at the time of HSE enrichment. We would rather refrain from speculation given these unknowns at this juncture. We note that crustal thickness estimates and their timing are strongly model-dependent.

- Oxygen isotopic abundances (specifically lines 143-145 plus subsequent discussion): in the manuscript it is stated that the oxygen isotopic composition of the three olivine-dominated

meteorites suggests affinities to other families of meteorites like brachinites and angrites. This seems a bit at odds with what is shown in Fig. 1B, where NWA 12319 and 12562 have values at the boundary between the high-end values of the oxygen-similar eucrites and the lower boundaries to the $\delta^{17}\text{O}$ values of oxygen-anomalous eucrites. NWA12217 falls instead between the angrites and the oxygen-anomalous eucrites (Scott et al. 2009; Zhang et al. 2019 and references within). I may be misunderstanding the plotted data but, given the general interpretation of the oxygen-anomalous HEDs and taking Fig. 1B at face value, it would seem that the olivine-dominated meteorites may come from at least two different bodies.

More O isotope data has been gathered and thus this section has been updated. NWA 12217 may indeed have O isotopes that are analogous to the anomalous HED meteorites and thus be sourced from a different, related parent body, assuming the canonical magma ocean models for Vesta's formation. Taken together with the Cr data, however, all three rocks are isotope-anomalous and could all be from different parent bodies.

Other comments

- Line 1 (Title), lines 25-27 (last sentence of the abstract/introductory paragraph), line 75: the manuscript links the discovery of the three olivine-dominated meteorites to the end of the “great dunite shortage”, i.e. the lack/paucity of olivine-dominated meteorites/asteroids.

The discovery of the three meteorites is a step in the direction of ending this open issue but stating that it marks its end is too definitive a statement, since we still have only a limited number of samples that the authors suggest being linked to only one body.

If the samples are all related to Vesta, they do mark an end to the lack of vestan mantle samples (another major open issue, as testified by the number of thematic studies linked to the Dawn mission), but in this case these sentences should be reframed in the specific vestan context.

As we can no longer conclude that these rocks all originate from Vesta, wording was changed to reflect the fact that these are likely related to Vestoids. The title declaration was also walked back.

- Lines 73-74: the reference to the Nice model is incorrect here: while early claims by the original series of papers argued for major effects on the asteroid belt, particularly in terms of excitation and ejections, later works (Minton & Malhotra 2009; Morbidelli et al. 2010; Pirani et al. 2016, with the latter also exploring the collisional implications) have clearly shown that the effects of the Nice Model on the asteroid belt are much more limited than originally claimed (basically, less than the combined effects of the subsequent 0.1-1 Gyr).

In terms of the Grand Tack: it can provide an explanation but its complex migration pattern is hardly required. Turrini et al. (2012) have shown that the formation of Jupiter is condition enough (migration is important but is not strictly required) for the collisional erosion of early formed planetesimals. Later works by Raymond & Izidoro (2017) and Pirani et al. (2019) confirmed, with more detailed disk models (and in the case of the latter more self-consistent migration scenarios), the natural emergence of the conditions to trigger the collisional erosion described by Turrini et al. (2012) without the need of any ad-hoc condition (like in the Grand

Tack).

Thank you. We have added a brief discussion of recent models.

- Lines 211-221: there are additional sites where olivine has been suggested by different authors (Ruesch et al. 2014; Nauthes et al. 2015; Palomba et al. 2015; see Turrini et al. 2016 for a global map), all associated with shallow craters. At the same time, the presence of olivine has been ruled out inside the two South Polar basins and on the central mount of Rheasilvia (Ammannito et al. 2013; Ruesch et al. 2014), which seems to indicate the lack of widespread olivine plutonic deposits over a range of depths and an area of the order of 25% the surface of Vesta.

This dichotomy is what brought Consolmagno et al. (2015) to point out the difficulties in quantitatively constraining the internal structure of Vesta and, tangentially, subsequent authors to suggest that the olivine detected on Vesta could have an exogenous origin (Nathues et al. 2015; Le Corre et al. 2015; Turrini et al. 2016). Can the serial magmatism in large magma chambers be reconciled with this dichotomy in the olivine distribution? Do the petrological and geochemical characteristics of the three meteorites provide additional insight on the issue of the distribution of olivine on Vesta?

The discussion was removed, as it seems less relevant now that we cannot argue that these rocks are necessarily Vestan.

Minor comments:

- Lines 34-36: one could make the point that the existence of iron meteorites belonging to tens of compositional groups, notwithstanding the depletion of the asteroid belt, observationally supports the theoretical argument.

Added sentence on iron meteorite diversity.

- this is mostly a matter of style, but in various points in the manuscript the adjective "Vestal" is used to mean "of Vesta". At the time of the arrival of Dawn to Vesta there has been quite a debate on the correct adjective to use, as "Vestal" has a very specific connotation (connected to the religious cult of the Roman goddess Vesta) and was therefore not adopted. The adjective "Vestan" was considered more correct and adopted, analogously to why we use "Martian" instead of "Martial" for Mars.

Changed "Vestal" to "Vestan".

Reviewer #2 (Remarks to the Author):

Review of "End of the Solar System Asteroidal Dunitic Shortage," by Z. Vaci et al.

The authors have presented an interesting work regarding remarkable olivine-rich achondrites, presenting new data regarding the petrography, bulk geochemistry, mineral chemistry, oxygen isotopes, rare earth elements, and highly siderophile elements. The authors make the case that the

meteorites have affinity with HED meteorites, and may thus represent a portion of the mantle of Vesta (with the caveat that they could originate from another differentiated parent body).

Overall, I find the manuscript to be well written, although some phrasing and grammatical changes are recommended below.

My biggest concern is with trying to parse the logic linking the title of the manuscript with the conclusions of the text. It may be that I have simply misunderstood an argument (in which case, I would recommend clarifying the text), but I am having trouble with the logic of the manuscript:

1) The title and final sentence of the abstract claim that these three meteorites represent an end to the dunite shortage of the Solar System.

2) The shortage of olivine-rich material is introduced as being based on the lack of observations of olivine rich asteroids or meteorites (although notably, there is a non-zero number of both).

3) Three new meteorites are presented and are shown to be similar to HED's, and especially Diogenites (although olivine rich instead of pyroxene rich), and other known possible parent bodies are ruled out.

4) The authors conclude with two possibilities:

A: The meteorites originate from an otherwise unknown object that was destroyed, thus favoring the idea that the missing mantle problem is solved via widespread destruction of such material ("Battered to bits" hypothesis).

B: The meteorites originate from Vesta, and most probably from Arruntia or Bellicia craters, where olivine rich material was excavated.

The conclusions follow from the analysis of the samples, but I am struggling to understand how either scenario A or B marks an "end" to the dunite shortage. In the case of A, it is basically a claim that an already proposed hypothesis is correct (although the Battered to bits hypothesis would not seem to be a unique explanation for these olivine-rich meteorites). In the case of B, it is a Vesta-specific claim that does not address the broader problem - this essentially argues that the deep mantle material of Vesta was never really ejected, but does not say anything about other planetesimals. Perhaps the authors are trying to claim that the plutonic origin of the meteorites, rather than a magma ocean origin, is responsible for much greater amounts of pyroxene rather than olivine crystallization? But in this case, it should be made more explicit.

Thank you for this important point. We have rewritten the conclusions substantially to reflect these and other criticisms. These rocks are not necessarily from Vesta, so scenario A seems to be likely. However, they are likely from similar parent bodies such as the Vestoids. The title was changed to reflect our interpretations.

I would like the authors to clarify the link between the title and the conclusions of the manuscript. This might also be aided with a summary diagram that explains the claims of the authors.

Detailed comments:

Title: As above, the "end" of the dunite shortage should be resolved, and as this study does not

involve asteroids at all, "Asteroidal" should probably be removed from the title to avoid confusion.

Title was changed to add nuance. As the study probably involves multiple parent bodies, asteroids are likely implicated.

Line 14-15: "The great dunitite shortage...represent the paucity of olivine rich asteroids" is not correct; please reword.

Reworded.

Line 22: "Consistent with being related to" - please rephrase.

Rephrased.

Line 23: The Cr-rich veins and symplectites are only really discussed in the supplemental information and not so much in the main text (barring a brief mention in lines 88-90). If this is a key line of evidence, please discuss it more in the main text. Especially so in context of exogenous fluid and melt interaction (which is described well in the supplement). If this is not critical, then it probably doesn't need to be in the abstract.

This is not critical to the narrative regarding origin of these meteorites, but the symplectites and veins do serve as diagnostic features that link the three rocks together. Removed from abstract.

Line 25-27: As mentioned above, this is an important and substantial claim, but the text does not seem to quite arrive at this conclusion.

Since these are olivine-rich breccias from multiple parent bodies, they do herald an end to the shortage of olivine-rich material. These are likely the first of many of these olivine-rich meteorites that have been missing from the collection.

Line 32: "Either" typically refers to precisely 2 items; please rephrase.

Deleted "either".

Line 35: The hyphens around "and meteorite samples of them" should be replaced with commas or parenthesis.

Replaced with commas.

Line 56: "...and complementarily there have not been..." Please rephrase.

Deleted “complementarily”.

Line 67-74: This simple explanation for the lack of olivine-rich meteorites seems fairly satisfactory. If widespread differentiation never took place, then there is not so much of a problem to be solved. Could you please clarify why the dunite shortage remains a problem?

The diversity of iron meteorites (note on this added earlier) and results of dynamical simulations (discussed earlier and later) suggest a large number of differentiated planetesimals and their associated olivine-rich mantles.

Line 102: I am uncertain what it means for a trace phase to be "well-distributed"? Please clarify. Also, is it correct that feldspar is a trace phase, is evenly(?) distributed, is present as brecciated fragments, and is associated with pure silica?

Rephrased sentences about feldspars.

Line 105: Feldspar as inclusions in olivine with radiating fractures might be an interesting image. Is this suggestive of a fluid inclusion?

The feldspars, especially K-feldspars and their associated silica-rich mesostasis, are likely suggestive of late-state melt inclusions. These should be studied further and probably warrant their own paper along with the symplectites. Added supplementary figure (S3).

Line 113-114: See comments on Figure 1.

The O isotope section and figure has been reworked.

Line 114: A reference to Figure 1B should be here.

Added.

Line 122: "(72)" was left in the PDF; please remove.

Removed.

Line 123: "Incompatible" should be lower case.

Changed.

Line 138-139: Please include a reference for how Pd can be added via terrestrial weathering.

Wording added and discussion is now given.

Line 140: "Subtracting the additions..." Please clarify: does this mean "Not including" or does this mean a mathematical adjustment to the data?

Changed to "not including".

Line 145-149: The argument being made here appears to be that angrites cannot be the source of the meteorites in this study due to oxygen fugacity; however, the meteorites are also thought to represent the deep mantle of their parent body, which would likely be more reducing than the shallower portions of the parent body. It is not immediately clear to me that the oxidation state of the meteorites in this case can distinguish the parent body. Can this be clarified?

While it is possible that more reduced angrites from the mantle of its parent body exist, the several other lines of evidence discussed in this paragraph, especially the fact that a plutonic, olivine-rich angrite has been identified and it bears no resemblance to the present meteorites, do not suggest affinity to the angrite parent body. We could make a similar argument about a hypothetical reduced mantle endmember of any other achondrite group, but this would not present a justification to ignore all of the other evidence to the contrary.

Line 156 and elsewhere: Please use "because" instead of "since."

Changed all non-temporal usages of "since" to "because".

Line 171-178: The comparison with diogenites and howardites does not mention oxygen fugacity, which was a primary listed reason for rejecting angrites.

Added note on fO_2 of ultramafic achondrites and HEDs.

Line 189: "the incorporation" is difficult to parse; please change the phrasing.

Changed to "crustal incorporation"

Line 187-192: This section can be made more compact by combining sentences and rephrasing. I am not certain that it is critical in any case, and this discussion of potential LREE enrichment could perhaps be expanded and migrated to the supplementary information; it seems like it would fit well with the symplectite discussion.

Reviewer #1 raised several points regarding the REE of these meteorites so we are keeping this discussion. Rephrased for succinctness and deleted reference to metasomatism.

Lines 194 and 199: The statement is made that HSEs are not sufficient for establishing a relationship with diogenites, but later in the paragraph, the similarity of the HSEs to diogenite

patterns are said to be suggestive of a relationship with Vesta. Please rephrase this paragraph to resolve this apparent contradiction.

Added a sentence on HSE evidence.

Line 208: In the paragraph explaining possible explanations for the dunite shortage (lines 54-66), the "Battered to Bits" hypothesis is not mentioned. Please discuss this earlier in the text.

It is discussed, but not named as such (line 59-60) and Burbine et al. (1996) is referenced.

Line 216: "...the study argues..." should be "...this study argues..."

Changed.

Line 217-219: This is interesting; can the relevance to the HEDs of plutonism as opposed to magma ocean be discussed?

This section has been rewritten.

Line 220-221: I take this to mean that the reason we don't have more dunitic material from Vesta is that it was never excavated and ejected?

Or it was ejected by impacts that we do not have meteoritic records of. We speculate on the reason Dawn did not observe mantle material in deep craters.

Line 222: A paragraph explaining how the dunite shortage is solved would be appreciated here.

This section has been rewritten. These meteorites represent the first set of olivine-rich samples, and there is no reason to assume there are not many more waiting to be found. This constitutes the end of the dunite shortage among meteorite samples. Had we not found these, one might assume there is a reason for a shortage among the meteorite record, but since these exist, the reason might just be not enough samples.

Line 225: How many thin sections were made?

Added number of thin sections.

Figure 1: Please label the phases in A. Images of the other two samples would also be appreciated, even if they are in the supplemental information. In B, it appears that you have plotted Iberita and A-881394 O isotopes. As eucrites, these samples are a bit unusual in their ^{17}O isotopic composition. Perhaps they should be labeled as outliers. Also, this study argues that the meteorites are most closely related specifically to diogenites and a subset of howardites; it would be good to plot the H, E, and D samples with distinct symbols. Further, the samples of

NWA 12217 appear to have significantly higher 17-O isotopic composition than the other samples. I cannot find a data table with the O isotopes in the submitted files to try to replot these data. This graph does not look consistent with the statement that the samples are indistinguishable. Is it correct to assume that the error bars associated with these data points are smaller than the symbols?

This figure has been reworked to incorporate additional data and commentary.

Figure 2: Please include a legend on this figure. The chondrite normalization is Anders and Grevesse, 1989; was there a reason this was used instead of a more recent reference, such as Lodders et al., 2009? B: I am accustomed to seeing the HSEs in order of vaporization; is there a reason for the order presented here?

The CI normalization was updated to Lodders et al. (2009). We order them to their order of incompatibility. The first is a nebula type of process, whereas the second follows Goldschmidt's classification which is much more relevant when examining igneous processes.

Figure 3: Are the Mg-rich Harzburgites the same as the Howardites? Please make this clear. As before, it would make sense to separate howardite, eucrite, and diogenite samples in this plot, as it is relevant to the details of the interpretation.

Since this graph shows olivine analyses, the HED are overwhelmingly diogenites. Mg-rich harzburgites are clasts found in howardites by Hahn et al. (2017). These notes were added to the description.

Supplementary information

Text appears to be fine.

It would be nice if additional thin section images could be shown here to support the textural interpretations made in the main text.

Added a supplementary figure (S10).

Figure S2: Sample NWA 12217 appears in this diagram, and also in several others, to plot far away from the other two. The 17-O isotopic composition is higher, the Mg# of olivine is higher, the modal percentage of olivine is higher, and the feldspars are quite distinct. It also has lower Sc, Cu, and REE abundances than the other two samples. Could you please include a summary of why the three samples all originate from the same parent body?

A note was added about how NWA 12217 could originate from a different parent body analogously to the oxygen anomalous HED meteorites. Description of differing habit of feldspar was added to S2. The O isotopic composition of NWA 12217 is an open question as with the oxygen-anomalous eucrites, but similarities in major and minor element composition of olivine and pyroxene, similar REE, textural similarities, identical symplectites and veins, and the difficulties posed by sampling biases via both the brecciation process and the selection of material to analyze suggest that 12217 is very much related to the other two. To take feldspar

compositions as an example, our inability to find more anorthite-rich clastic feldspars in NWA 12217 does not rule out their existence, and likewise with our inability to find albitic or alkali feldspars in NWA 12319 or 12562. As this question of parent bodies cannot be answered using our data set, we think it best to leave this open.

Figure S6-S7: These images are beautiful.
Agreed.

Reviewer #3 (Remarks to the Author):

Dear Dr. Vaci et al.,

Here you will find my review of the manuscript “End of the Solar System asteroidal dunite shortage”.

Identifying and characterizing meteoritic material that may represent the mantle of differentiated asteroids is an important endeavor that will provide insights into one of the earliest stages of asteroid differentiation. The manuscript shows that 'missing' dunites have been identified and reports their characterization in detail. While that itself is interesting, the manuscript convincingly argues that the studied samples (NWA 12217, 12319, and 12562) may be from asteroid Vesta, adding significant weight to the importance of this work.

The manuscript is well-written, well-presented, and was a pleasure to read. I thought it laid out the so-called ‘dunite shortage’ issue well and made a convincing argument that such samples have now been found. I believe that this work will be of substantial interest to the extraterrestrial sample science community. I have no hesitation recommending this work for publication in Nature Communications after minor revisions.

Suggestions that should be addressed, include:

1. Line 14: Introduce what a dunite is for the non-expert reader.

Added definition of dunite.

2. Line 118/Fig. 1B: These samples do not seem consistent with brachinites. HEDs/angrites? Sure. But they do not plot with the brachinites. I suggest cutting mention of brachinites here.

The range in brachinite O isotopic composition is large enough to warrant plotting them in the figure (note the two samples in the bottom left corner). Unlike the HED, the majority of which define a fairly tight fractionation line (see updated plot), the large scatter in brachinite O isotope compositions allows for potential association with the ultramafic achondrites.

3. Line 122: Remove struck through parentheses.

Removed.

4. Fig. 2: Add legend for symbols to graph; this is easier for the reader than having to dig through the caption.

Added.

5. Line 144: Once again, the O-isotopes of these samples do not seem consistent with brachinites. This supports your later conclusion that these samples are not related to brachinites.

See response to suggestion 2.

6. Supplementary data set: some of the microprobe data are of poor quality (totals <98 wt.%) and should not be reported. It appears that many (but not all) of these analyses have been flagged as poor quality in the annex but I suggest removing them from the data file and all relevant figures. They should also be removed from the calculations of Mg# if they were included. Furthermore, cations are not reported for any of these analyses; stoichiometry of all analyses should be calculated to check the quality of the reported data prior to publication.

Low totals were removed, with the exception of chromite. The chromite low totals are possibly due to additional cations such as V that were not analyzed. All phases are stoichiometric.

I look forward to seeing this manuscript in print.

REVIEWERS' COMMENTS

Reviewer #1 (Remarks to the Author):

The revised manuscript submitted by the authors has been updated with new results from additional analyses. The updated description of the three meteorites as well as the discussion of their similarities and differences, both among themselves and with other differentiated meteorites, presents a complex and fascinating picture. I agree with the authors that there is no reason not to expect further and more frequent finds of similar meteorites in the future, and that their discovery shed a different light on the missing mantle problem.

The major comments I have focus on the link between the three meteorites studied in this work and Vesta/HEDs:

- lines 212-215: the sentences "While the eucrites and diogenites, as crustal rocks, show LREE enrichments relative to the HREE, residues of partial melting are expected to have LREE/HREE < 1. Instead, NWA 12217 and 12562 show enrichments in the LREE relative to the HREE..." appear at odds with fig 3A. In the figure diogenites show LREE < HREE, i.e. they have the opposite trend than NWA 12217 and 12562, while the main text states they should have the same trend.

- lines 231-232: "Their similarity to diogenites and Mg-rich harzburgites suggests that these parent bodies are sampled by V-type asteroids or vestoids." -> The statement as it is formulated is partially incorrect as the majority of V-type asteroids are dynamically linked to Vesta, with recent dynamical (Spoto et al. 2015) and geologic (Schenk et al. 2012) evidences linking them to the formation of Rheasilvia and Veneneia. As none of the three meteorites fully match the characteristics of the HEDs, they are more likely to sample only V-type asteroids not dynamically linked to Vesta, not all V-type (otherwise they would still be sampling Vesta).

- lines 239-241: "Coupled with the oxygen and chromium isotopic evidence, this suggests the existence of at least two large, differentiated parent bodies present in the early Solar System, at least one of which has since been annihilated by impact (the other potentially being Vesta)." -> the results of the authors seem to actually suggest that the three meteorites originated from at least two bodies similar yet distinct from Vesta, as none of them fit the pattern of HEDs meteorites in terms of O and Cr composition.

Other comments:

- line 30: "differentiated planets" should probably be "differentiated planetary body"

- lines 171-172: the text still states "The oxygen isotopic compositions [...] suggest affinities with angrites, brachinites and [...]", yet from Fig. 1B one can see some affinity with angrites and anomalous HEDs but there is no discernible affinity with brachinites. The O-Cr analysis further distances the three samples from angrites and brachinites (as also the authors point out in their rebuttal), so this sentence should be revised.

- line 249: "complimentary" -> "complementary" (complimentary is "free")

- lines 356-357: "with highly positive $\epsilon^{54}\text{Cr}$ " is repeated twice

Reviewer #2 (Remarks to the Author):

The authors have revised the manuscript, and the revisions are fairly satisfactory. This is an interesting piece of work, and one that I would be glad to see published. The revised title is more

appropriate, although the authors might consider adding the phrase, "Newly discovered." I would prefer if hand sample images of the meteorites could be included in the supplement. There is additional work on Figure 2 that must be completed before publication.

Line 15: I would rearrange this sentence to put "Mantles of rocky planets..." at the beginning so that the first word of the manuscript is not "seismic."

Line 27: "End to the shortage of olivine-rich material" -

Line 79: "in-situ" should be "in situ" and italicized.

Line 118: Up to 25% - what is the lower bound?

Paragraph starting on line 245: This is interesting information, but it seems disconnected somehow. It is unclear to me what you are trying to argue for here. Please rephrase/connect more directly with the current study.

Figure 2: The text in all four boxes is much too small. The panels are named in the format of "(A)", but figures 1 and 3 panels are formatted as "A" only. Please homogenize.

A) It is unclear to me why circles were chosen for both the samples in this study and the literature data. The caption repeats information from the legend - this can be omitted from the caption. I am unclear on the UC Davis data - is that not from this study? The white fill on those symbols makes them confusing, because it distracts from the main samples in this study. The labels of the samples from this study should be expanded so that each of them is given by name in the legend. The blue bubble is labeled both "Earth" and "Earth, ECs, and aubrites." The colorful labels of the main samples are confusing, as this is already a tremendously colorful figure.

B) The colorful bubbles are not labeled. The colored labels of the key samples are difficult to find because of the colors. There are white squares and filled squares - are these both eucrites? Unclear from the legend. There is a label in the upper blue field that is impossible to read because of the data plotted on top of it.

C) It is unclear what the point of this panel is. Of course these are not carbonaceous chondrites. Perhaps this can be moved to the supplement?

D) Circles are used for both howardites and samples from this study.

Dr. Matthew S. Huber
University of the Western Cape, South Africa

Reviewer #3 (Remarks to the Author):

Dear Dr. Vaci et al.,

Here you will find my review of the revised manuscript "End of the Solar System asteroidal dunite shortage".

Identifying and characterizing meteoritic material that may represent the mantle of differentiated asteroids is an important endeavor that will provide insights into one of the earliest stages of asteroid differentiation. The manuscript shows that missing dunites have been identified and reports their characterization. The work is very interesting and certainly worthy of publication.

The first round of reviews appears to have been mostly addressed. However, the original manuscript overstated the impact of the proposed work and this has not been entirely remedied in the revised version of the manuscript. For example, see the last sentence of the abstract. Furthermore, the new conclusion that the studied samples are not from Vesta, while convincing based on the new data, significantly decreases the potential impact of the manuscript.

REVIEWERS' COMMENTS

Reviewer #1 (Remarks to the Author):

The revised manuscript submitted by the authors has been updated with new results from additional analyses. The updated description of the three meteorites as well as the discussion of their similarities and differences, both among themselves and with other differentiated meteorites, presents a complex and fascinating picture. I agree with the authors that there is no reason not to expect further and more frequent finds of similar meteorites in the future, and that their discovery shed a different light on the missing mantle problem.

The major comments I have focus on the link between the three meteorites studied in this work and Vesta/HEDs:

- lines 212-215: the sentences "While the eucrites and diogenites, as crustal rocks, show LREE enrichments relative to the HREE, residues of partial melting are expected to have LREE/HREE < 1. Instead, NWA 12217 and 12562 show enrichments in the LREE relative to the HREE..." appear at odds with fig 3A. In the figure diogenites show LREE < HREE, i.e. they have the opposite trend than NWA 12217 and 12562, while the main text states they should have the same trend.

Thank you! After carefully re-examining the REE systematics of the eucrites and diogenites, there does not seem to be a tendency for LREE enrichment in the eucrites. Instead there is a large amount of variability, as with the REE systematics of the diogenites. So I removed this comparison with HED.

- lines 231-232: "Their similarity to diogenites and Mg-rich harzburgites suggests that these parent bodies are sampled by V-type asteroids or vestoids." -> The statement as it is formulated is partially incorrect as the majority of V-type asteroids are dynamically linked to Vesta, with recent dynamical (Spoto et al. 2015) and geologic (Schenk et al. 2012) evidences linking them to the formation of Rheasilvia and Veneneia. As none of the three meteorites fully match the characteristics of the HEDs, they are more likely to sample only V-type asteroids not dynamically linked to Vesta, not all V-type (otherwise they would still be sampling Vesta).

Actually, new data that we have updated with this revision does make all three meteorites strongly connected with the bona fide HEDs, hence Vesta, or V-type asteroids that are dynamically linked to Vesta. So the statement would remain true. (see our revised Fig. 2, data table and the updated/revised main text. Our earlier data appeared to have suffered from low Cr yield. We have recalibrated all three columns used in our Cr extraction protocol and recalculated the full yields and documented to this effect for the three ultramafic samples in the Method section.

- lines 239-241: "Coupled with the oxygen and chromium isotopic evidence, this suggests the existence of at least two large, differentiated parent bodies present in the early Solar System, at least one of which has since been annihilated by impact (the other potentially being Vesta)." -> the results of the authors seem to actually suggest that the three meteorites originated from at least two bodies similar yet distinct from Vesta, as none of them fit the pattern of HEDs

meteorites in terms of O and Cr composition.

A strong case can now be made to connect these three ultramafic meteorites with Vesta (normal HEDs) with the new data obtained for the revision (see our new Fig. 2, the new isotope data table [Table S5], and the revised main text). See explanations above and excel sheet calculations and schematics illustrating how the “rotational effect” might have generated artifacts in the low yield chemistry in our earlier data.

Other comments:

- line 30: "differentiated planets" should probably be "differentiated planetary body"

Changed.

- lines 171-172: the text still states "The oxygen isotopic compositions [...] suggest affinities with angrites, brachinites and [...]", yet from Fig. 1B one can see some affinity with angrites and anomalous HEDs but there is no discernible affinity with brachinites. The O-Cr analysis further distances the three samples from angrites and brachinites (as also the authors point out in their rebuttal), so this sentence should be revised.

Revised sentence to emphasize lack of association with angrites and brachinites.

- line 249: "complimentary" -> "complementary" (complimentary is "free")

Changed.

- lines 356-357: "with highly positive $\epsilon^{54}\text{Cr}$ " is repeated twice

Removed.

Reviewer #2 (Remarks to the Author):

The authors have revised the manuscript, and the revisions are fairly satisfactory. This is an interesting piece of work, and one that I would be glad to see published. The revised title is more appropriate, although the authors might consider adding the phrase, "Newly discovered." I would prefer if hand sample images of the meteorites could be included in the supplement.

Added hand sample images to the supplement. We disagree with the necessity of “newly discovered” as we believe this is explicitly stated in the paper. In time, the modifier in the title will be defunct.

Based on the newly revised Cr isotope data (see our new Fig. 2), combined with oxygen isotope data, stronger case can now be made with the link to Vesta for these three ultramafic rocks (see above for our explanation to reviewer 1). Hence a new updated title.

There is additional work on Figure 2 that must be completed before publication.

Figure 2 was revised (see below) with newly obtained data.

Line 15: I would rearrange this sentence to put "Mantles of rocky planets..." at the beginning so that the first word of the manuscript is not "seismic."

The sentence was rearranged.

Line 27: "End to the shortage of olivine-rich material" –

I am not sure what this comment is asking for.

Line 79: "in-situ" should be "in situ" and italicized.

Changed.

Line 118: Up to 25% - what is the lower bound?

The lower bound is 0. Specified this.

Paragraph starting on line 245: This is interesting information, but it seems disconnected somehow. It is unclear to me what you are trying to argue for here. Please rephrase/connect more directly with the current study.

We brought this up because several reviewers wanted some discussion of Dawn data, and the lack of evidence for a large mantle on Vesta or the presence of olivine is relevant to the existence of these olivine-rich meteorites.

Figure 2: The text in all four boxes is much too small.

Font size increased.

The panels are named in the format of "(A)", but figures 1 and 3 panels are formatted as "A" only. Please homogenize.

Homogenized

A) It is unclear to me why circles were chosen for both the samples in this study and the literature data.

These are legacy symbols. It is now cleaned up.

The caption repeats information from the legend - this can be omitted from the caption. I am unclear on the UC Davis data - is that not from this study?

They are not from this study.

The white fill on those symbols makes them confusing, because it distracts from the main samples in this study. The labels of the samples from this study should be expanded so that each of them is given by name in the legend.

Done

The blue bubble is labeled both "Earth" and "Earth, ECs, and aubrites."

Only "Earth, ECs, and aubrites" is kept

The colorful labels of the main samples are confusing, as this is already a tremendously colorful figure.

This is necessary to distinguish different groups.

B) The colorful bubbles are not labeled.

They are properly labeled now.

The colored labels of the key samples are difficult to find because of the colors. There are white squares and filled squares - are these both eucrites? Unclear from the legend.

One is for anomalous eucrites, another for normal eucrites

There is a label in the upper blue field that is impossible to read because of the data plotted on top of it.

This is now corrected.

C) It is unclear what the point of this panel is. Of course these are not carbonaceous chondrites. Perhaps this can be moved to the supplement?

This is to show that mixing exogenous carbonaceous material with main group HED cannot reproduce the compositions of the ultramafic achondrites. But one of the three samples, NWA 12217, with higher $\Delta^{17}\text{O}$, could be explained by mixing with an ordinary chondrite component. This is explained in the revised main text. Without panel C, panel D has no meaning. We cannot remove panel C and just keep panel D as the reviewer suggested. They support and depend on each other. It is critically important to show that the anomalous composition for NWA 12217 could be a result of mixing of normal HEDs (Vesta) with an ordinary chondritic component.

D) Circles are used for both howardites and samples from this study.

Resolved in the revised figure

Dr. Matthew S. Huber
University of the Western Cape, South Africa

Reviewer #3 (Remarks to the Author):

Dear Dr. Vaci et al.,

Here you will find my review of the revised manuscript “End of the Solar System asteroidal dunite shortage”.

Identifying and characterizing meteoritic material that may represent the mantle of differentiated asteroids is an important endeavor that will provide insights into one of the earliest stages of asteroid differentiation. The manuscript shows that missing dunites have been identified and reports their characterization. The work is very interesting and certainly worthy of publication.

The first round of reviews appears to have been mostly addressed. However, the original manuscript overstated the impact of the proposed work and this has not been entirely remedied in the revised version of the manuscript. For example, see the last sentence of the abstract.

Softened the wording of the last sentence in the abstract.

Furthermore, the new conclusion that the studied samples are not from Vesta, while convincing based on the new data, significantly decreases the potential impact of the manuscript.

Our revised and updated new data with fuller Cr recovery yield support a much stronger bona fide normal HED connection, hence a Vesta origin for these three ultramafic rocks. Low Cr yield, resulting in a mass fractionation on the column that deviates from the kinetic exponential law used in all mass spectrometry, could generate artifacts in mass independent anomalies, a fact that is widely known and must have been the case in our earlier results. We are glad that this issue is now resolved in the last minute before publication, and we are able to present the final, accurate data with confidence. See attached excel spreadsheet showing the “rotational effect” comparing the old and new data due to the loss of Cr to the tail end of column, resulting enrichment of light isotopes (^{50}Cr), and how it would “rotate” the rest of the heavy isotopes when not fully collected and the exponential law used in mass spectrometry could not capture this and correct it effectively.